# BI-PHASE TRAINING
# LEARNING EFFICIENTLY IN HIGH-DIMENSIONS

## ABSTRACT

Pre-trained foundation models have achieved remarkable generalization across a wide spectrum of downstream tasks. However, as models scale in size, the cost to pre-train models becomes prohibitively expensive. In this work, we introduce Bi-Phase Training (BPT), a novel parameter-efficient pre-training method designed to capture the expressiveness of fully parameterized models while drastically reducing the number of trainable parameters. BPT achieves this by combining constrained high-rank transformations using diagonal matrices with exploration of lower-dimensional subspaces through low-rank matrices, facilitating effective optimization within a reduced parameter space. We empirically demonstrate the effectiveness of BPT across various model scales, showing that it successfully matches the performance of standard pre-training on language models while achieving significant reductions in trainable parameters, such as a 66% reduction of trainable parameters for a 1.5B model. Furthermore, we conducted a comprehensive evaluation of 17 diverse downstream tasks, confirming that models trained with BPT maintained performance comparable to those trained with a fully parameterized standard method.

## 1 INTRODUCTION

Foundational models have demonstrated impressive general purpose performance(DeepSeek-AI et al., 2025; Grattafiori et al., 2024). These models usually consist of billions of parameters, are trained on massive datasets, and have become the standard in modern deep learning. Empirical studies such as scaling laws have demonstrated that increasing model size generally leads to lower training loss (Kaplan et al., 2020; Hoffmann et al., 2022), while also giving rise to emergent behaviors, including complex reasoning abilities, that only appear beyond certain scale thresholds.

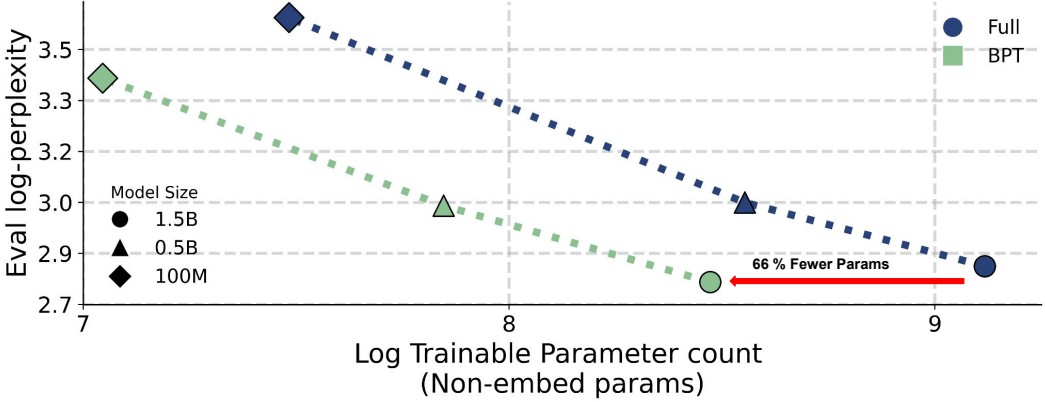

Figure 1: Evaluation log-perplexity vs. log trainable parameters (non-embedding) across model sizes. BPT matches the fully-parameterized baseline while using far fewer trainable parameters; on the 1.5B model it achieves comparable eval loss with 66% fewer trainable parameters

Despite these advancements, the continued scaling of foundation models faces practical limitations. Although there has been substantial progress from a systems perspective, ranging from more efficient attention mechanisms (Dao et al., 2022) to low-precision training techniques (Peng et al., 2023; Kalamkar et al., 2019) and innovations in model architectures such as the widespread adoption of Transformers (Vaswani et al., 2023), Mixture-of-Experts (MoE) (Shazeer et al., 2017), and state-space models (Gu et al., 2022). However, the dominant training paradigm, full parameter optimization, remains largely unchanged.

The scaling law (Kaplan et al., 2020; Hoffmann et al., 2022) roughly states that as we increase the number of parameters $N$, we should observe our loss $L$ to decrease propositional to a power law of the form:

$$L \propto N^{-\alpha} \tag{1}$$

for some positive constant $\alpha$, typically estimated empirically. This relationship suggests that larger models tend to perform better, provided sufficient data and compute resources (Hoffmann et al., 2022). However, this relation in Equation 1 only depends on the total number of parameters rather than the **total trainable parameters**. This distinction is critical, as each trainable parameter incurs a fixed memory overhead: the parameter itself ($n$), its gradient ($n$), and two optimizer states ($2n$) when using Adam family of optimizers (Kingma & Ba, 2017; Loshchilov & Hutter, 2019). While total training memory depends on several factors such as batch size and activation footprint (Chen et al., 2016), this $4n$ overhead remains fixed per trainable parameter. In this paper, we distinguish non-trainable and trainable parameters for pre-training, and interpret the total number of parameters as the model's **capacity to learn**. Our key claim is that models can learn as effectively with far fewer trainable parameters than the total number of parameters, as long as the learning dynamics are preserved.

To preserve learning dynamics while being parameter efficient, we introduce Bi-Phase Training (BPT), a novel parameter-efficient method for training foundation models that updates a weight matrix W using a combination of constrained high-rank and low-rank updates simultaneously.

Pre-training foundation models are computationally expensive, so every efficiency gained translates to substantial resource savings. Empirically, we demonstrate the effectiveness of our method by pre-training three language models of varying sizes, showing consistent performance across scales. We also perform a comprehensive downstream evaluation on our trained model to show that there is no loss of generalization between using fewer trainable parameters and all trainable parameters.

Our main contributions are as follows:

- We introduce Bi-Phase Training (BPT) to significantly reduce the number of trainable parameters required during pre-training.

- We provide a theoretical upper-bound to the update induced by BPT and provide empirical evidence to show that this method works across the model scales.

- We comprehensively test the method on 17 downstream evaluation tasks and show that it matches the performance of the fully parameterized model.

## 2 BACKGROUND

During neural network training, we aim to find a weight matrix $W \in \mathbb{R}^{n \times m}$ that minimizes a loss function, $L$. This matrix defines a linear transformation from an input space $\mathbb{R}^m$ to an output space $\mathbb{R}^n$. The matrix is updated iteratively using gradient signals from backpropagation. The rank of the update serves as a proxy for the number of independent directions explored during optimization, higher-rank updates enable more expressive subspace traversal better utilizing redundant dimensions. Formally, we want to optimize the weight matrix W using $\Delta W$:

$$W' = W + \Delta W$$

where $W'$ is the optimized weight matrix. For a single training example with an input vector $x \in \mathbb{R}^m$ and a corresponding layer output $y \in \mathbb{R}^n$, the gradient of the loss with respect to the weight matrix $G$ is the outer product of the upstream gradient and the input vector's transpose:

$$G = \frac{\partial L}{\partial W} = \left( \frac{\partial L}{\partial y} \right) x^T$$

Since this is an outer product of two vectors, its rank is at most 1. In stochastic gradient descent (SGD), where only the gradient of the matrix is used for the update, $\Delta W$, the rank of $\Delta W$ for a single example is at most 1. For a mini-batch of $B$ training examples. We can stack the inputs into a matrix $X \in \mathbb{R}^{m \times B}$ and the corresponding upstream gradients into a matrix $\Lambda \in \mathbb{R}^{n \times B}$. The gradient for the entire mini-batch is then given by the matrix product:

$$G = \Lambda X^T$$

The rank of a matrix product is less than or equal to the minimum rank of its factors. Since the rank of $\Lambda$ is at most $\min(n, B)$ and the rank of $X^T$ is at most $\min(B, m)$, the rank of the resulting gradient matrix is bounded by:

$$\mathrm{rank}(G) \le \min(\mathrm{rank}(\Lambda), \mathrm{rank}(X^T)) \le \min(n, m, B)$$

For the sake of simplicity, let $B \le \min(n, m)$. For the update $\Delta W$ in SGD, the $\mathrm{rank}(W)$ is at most $B$. In modern optimizers such as Adam (Kingma & Ba, 2017), where exponential moving averages (EMA) accumulate gradients overtime, at any timestep $t$, the rank of the update is no longer upper bounded by just $B$ but rather $\min(n, m, t \cdot B)$.

To practically reduce the number of trainable parameters in $W$, we ideally want to express $\Delta W$ with minimal number of parameters while having high-rank such that the trainable parameter reduction does not alter it's learning dynamics.

In LoRA(Hu et al., 2021), $\Delta W$ is defined to be product of two low-rank matrices $B \in \mathbb{R}^{n \times r}$ and $A \in \mathbb{R}^{r \times m}$ such that $\Delta W = \alpha BA$, where $\alpha$ is some scaling factor. LoRA yields strong results for fine-tuning a pre-trained weight, however, the objective landscape for pre-training from scratch is too complex to be modeled by $r$-dimensional subspace alone (Lialin et al., 2023). Since the $\Delta W$ rank in LoRA is always bounded by $r$, it can never utilize the full rank potential of $W$. To address this limitation, ReLoRA (Lialin et al., 2023) extends LoRA by accumulating low-rank updates over multiple steps:

$$\Delta W = s \sum_{n=1}^{N} BA$$

where B and A are low-rank matrices. During training, B and A are optimized, and their low-rank matrices are periodically merged back into the original weight matrix W. Following each merge, B and A are reinitialized, allowing the accumulated update $\Delta W$ to surpass the rank constraint $r$ after multiple merge cycles. Despite this improvement, the rank of updates at **any single step** remains confined to a low-rank subspace. Also for the duration of $\Delta t$ between two merge cycles, the update is restricted to only rank-$r$ subspace. This limits it's ability to learn as efficiently as fully parameterized model as the update rank for any given timestep can be high-rank and results in weaker performance compared to full parameter pre-training (Lialin et al., 2023). In a related work, HyperAdapt (Gurung & Campbell, 2025) proposed a parameter-efficient high-rank adaptation method by defining the update as:

$$\Delta W = AWB - W$$

where $A \in \mathbb{R}^{n \times n}$ and $B \in \mathbb{R}^{m \times m}$ are trainable diagonal matrices, and W is non-trainable parameter. Since both trainable matrices are diagonal, only $n + m$ number of parameters are needed to train them. Here the rank of $\Delta W$ is upper-bounded by the $2 \cdot \mathrm{rank}(W)$. However, this method assumes that pre-trained weight matrices already contain relevant orthogonal directions which can be scaled relevant to the downstream fine-tuning task, strictly limiting its application to fine-tuning pre-trained models.

## 3 OUR METHOD

The success of large foundation models, often linked to scaling laws (Kaplan et al., 2020; Hoffmann et al., 2022), suggests that models benefit significantly from a vast parameter space. For a model with a total number of parameters $C$, it represents the model's capacity to learn for a given architecture. We hypothesize that for the same model, only using the total number of trainable parameters $N$, where $N \ll C$, can approach the learning dynamics of a fully trainable model by matching the rank and geometry of its updates. In particular, since a fully parameterized model can realize high-rank

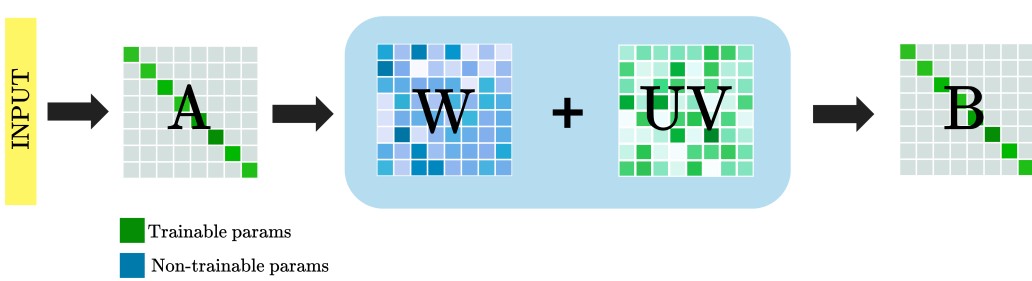

Figure 2: Bi-Phase Training Forward Pass: Here the green squares are trainable parameters in the model while the blue squares are frozen during training.

updates at any step $t$, we seek an update $\Delta W$ whose rank is comparably high while minimizing total number of trainable parameter to express $\Delta W$. To do this, we introduce Bi-Phase Training (BPT), a parameter-efficient method designed to induce high-rank updates within individual weight matrices using diagonal matrices and build low-rank subspace using low-rank matrices.

Our method conceptualizes the optimization of a weight matrix $W \in \mathbb{R}^{n \times m}$ not as a direct update to all $n \times m$ parameters, but as an update structure $\Delta W$ that combines transformations applied through efficient high-rank diagonal matrices with exploration guided by low-rank matrices. This allows us to approximate the expressiveness of full parameter updates while drastically reducing the trainable parameter count. Formally, for a weight matrix $W$, the update $\Delta W$ is defined as:

$$\Delta W = A \left( W + \sum_{t=1}^{T} UV \right) B - W \tag{2}$$

where $A \in \mathbb{R}^{n \times n}$ and $B \in \mathbb{R}^{m \times m}$ are diagonal trainable matrices, while $U \in \mathbb{R}^{n \times r}$ and $V \in \mathbb{R}^{r \times m}$ are low-rank trainable matrices. Also, $W$ remains fixed throughout training, meaning it does not receive a gradient update. At a given time $t$, we only keep a pair of U and V trainable and periodically merge U and V to W. So that means for a given step $t$, we have:

$$\Delta W = A \left( W + UV \right) B - W \tag{3}$$

We show that for any given time $t$, this update (Equation 3) is a high-rank update Theorem 3.1. Intuitively, the low-rank matrices U and V find and accumulate relevant orthogonal directions within a lower-dimensional subspace, while the diagonal matrices A and B scale relevant orthogonal directions, thus effectively guiding the high-rank updates. For a given input $x$, our modified forward pass becomes:

$$h = (W + \Delta W)x$$
$$= (A \left( W + UV \right) B) x$$

This forwards pass yields high-rank update using minimal trainable parameters.

**Lemma 3.1.** *The upper bound for the rank of the update* $\Delta W = A(W + UV)B - W$ *is* $2R + r$, *where* $R = \text{rank}(W)$ *and* $r = \text{rank}(UV)$.

*Proof.* We have
$$\Delta W = A(W + UV)B - W = AWB + AUVB - W.$$
For all conformable matrices X and Y, $\text{rank}(X + Y) \leq \text{rank}(X) + \text{rank}(Y)$. Therefore,
$$\text{rank}(AWB + AUVB - W) \leq \text{rank}(AWB) + \text{rank}(AUVB) + \text{rank}(-W).$$
Since matrix multiplication cannot increase rank, $\text{rank}(AXB) \leq \text{rank}(X)$ for any X, and hence
$$\text{rank}(AWB + AUVB - W) \leq \text{rank}(W) + \text{rank}(UV) + \text{rank}(-W).$$
We have $\text{rank}(UV) = r$ and $\text{rank}(W) = \text{rank}(-W) = R$, so
$$\text{rank}(\Delta W) \leq R + r + R = 2R + r.$$
Hence the rank of the update $\Delta W$ is upper-bounded by $2R + r$. Trivially, it is also bounded by its dimensions so $\Delta W \leq \min(n, m, 2R + r)$ $\qquad \square$

## 3.1 COMPOUNDING EFFECT

A key property of BPT is that the low–rank matrices $(U, V)$ and the diagonal matrices $(A, B)$ reinforce each other over time. Each merge step ( Equation 4) stores the currently discovered low–dimensional subspace into the base weight,

$$W := W + UV,$$

so that subsequent optimization starts from a matrix which contains the subspace discovered in the previous merge cycle by $UV$. Because $A$ and $B$ are *not* merged or reinitialized, they continuously rescale these accumulated directions, effectively bootstrapping the previously discovered subspaces while the next $(U, V)$ pair explores new ones ( Equation 2). This yields a high–rank update at each step with only a small number of trainable parameters.

Concretely, after $t$ merge cycles the span of $W$ contains the union of the $t$ discovered rank-$r$ subspaces, while the diagonal matrices $A \in \mathbb{R}^{n \times n}$ and $B \in \mathbb{R}^{m \times m}$ provide elementwise left/right scaling that can immediately modulate any of these directions without re-discovering them through another low–rank matrices $U$ and $V$. The result is a compounding effect: (i) low–rank factors $(U, V)$ *add* new useful directions to $W$; (ii) diagonals $(A, B)$ *amplify or attenuate* both old and newly added directions; and (iii) the process repeats from a progressively better-conditioned base.

This differs from ReLoRA–style training, where the update is always confined to a rank-$r$ subspace for the duration of $\Delta t$ between two merge cycles. Once merged, those directions become part of $W$ but are not actively and continuously reweighted. Consequently, ReLoRA must trade off between modifying previously learned directions and discovering new ones within a fixed rank budget at each step, whereas BPT can simultaneously (a) cheaply explore new subspaces via $(U, V)$ and (b) flexibly reshape all accumulated directions through $(A, B)$.

## 3.2 PARAMETER EFFICIENCY AND COMPUTATIONAL ADVANTAGE

The primary advantage of BPT lies in its significant reduction in the number of trainable parameters compared to training the full weight matrix $W$. For a single layer with weight matrix $W \in \mathbb{R}^{n \times m}$, the trainable parameters in BPT are located only in the diagonal matrices $A$ and $B$, and the low-rank matrices $U$ and $V$.

The total number of trainable parameters for one such layer in BPT is $n + m$ (from diagonal $A$ and $B$) plus $n \times r + r \times m$ (from low-rank $U$ and $V$). In contrast, training the full weight matrix requires $n \times m$ parameters. Since $r \ll \min(n, m)$, the total trainable parameters in BPT are substantially less than $nm$.

$$\underbrace{n + m}_{\text{Diagonal Matrices}} + \underbrace{nr + rm}_{\text{Low-Rank Matrices}} \ll \underbrace{nm}_{\text{Full Matrix}}$$

This parameter reduction directly translates into lower memory overhead during training with efficient kernel implementation. This trainable parameter count reduces the fixed memory cost associated with training neural networks as stated in section 1.

## 3.3 INITIALIZATION

Proper initialization is crucial for stable training. We initialize the fixed weight matrix $W$ using the standard Kaiming initialization (He et al., 2015). The diagonal matrices $A$ and $B$ are initialized to one, such that the diagonal matrices start as identity matrices. For the low-rank matrices $U$ and $V$, we initialize $U$ to be semi-orthogonal and $V$ to be zero. This means that $U$ has full rank for its capacity and also no direction is dominant. Since $V$ is zero, the product $UV$ also zeros out, so the initial forward pass would be the same as Kaiming initialization.

## 3.4 RE-INITIALIZATION AND MERGING

The optimization process in BPT involves learning the diagonal scaling factors in $A$ and $B$ and accumulating updates in the low-rank term $UV$. Periodically, we merge the learned low-rank subspace into the base weight matrix $W$. This process stores the progress made in navigating the parameter space and allows the low-rank components to explore new low-rank subspace. The merging

step updates the fixed weight matrix W:

$$W := W + UV \qquad (4)$$

After merging, the low-rank matrices U and V are re-initialized to be semi-orthongal and zero matrix respectively :

$$U := \text{init.orthogonal} \quad , \quad V := \text{init.zero}$$

Similar to initialization, the product UV zeros out, ensuring that re-initialization does not introduce a sudden change in the weight matrix used in the forward pass immediately after merging. This guarantees that the effective weight matrix $W + UV$ is equivalent to the weight matrix W from Equation 4 at the point of re-initialization. The random semi-orthongal initialization of U upon re-initialization ensures that it starts with a rank of $r$, ready to explore new low-dimensional subspaces effectively. Unlike the low-rank matrices, which are merged and re-initialized to explore new subspaces, merging the diagonal matrices does not yield exploration of new parameter subspaces.

Finally, merging and re-initializing the low-rank matrices makes the exponential moving averages(EMA) stored by the optimizer obsolete. However, since the diagonals are not merged, the EMAs remain intact, helping the training be stable without sudden loss spikes.

## 4 RELATED WORK

While the past few years have witnessed remarkable progress in reducing the memory footprint for fine-tuning large language models, memory-efficient pre-training remains considerably less explored. GaLore (Zhao et al., 2024) addresses this gap by proposing a novel approach to parameter-efficient training that projects gradients into a low-rank subspace, diverging from traditional methods that directly parameterize low-rank weight matrices. However, GaLore's reliance on Singular Value Decomposition (SVD) to identify the optimal low-rank approximation presents a significant bottleneck. The computational cost of SVD scales poorly with the dimensions of the matrix, making GaLore difficult to scale.

LoRA-the-explorer (LTE) (Huh et al., 2024) extends LoRA to pre-training by exploring the low-rank solution space in parallel across different computing nodes. While LTE's parallel search strategy presents a seemingly promising avenue to mitigate the computational bottlenecks associated with full-rank pre-training, it unfortunately inherits the same fundamental drawbacks, which is that at each step, optimization is only limited to a low-rank subspace similar to other low-rank methods section 2.

## 5 EMPIRICAL EXPERIMENTS

To evaluate the effectiveness of our proposed method, we conducted a series of pre-training experiments on language models of varying scales and assessed their performance on a diverse suite of downstream tasks. Our primary goal was to demonstrate that BPT can match the performance of standard full parameter pre-training while requiring significantly fewer trainable parameters, thereby lowering the computational barrier to developing large foundation models.

All models were pre-trained from **scratch** using the standard next-token prediction objective. We used a 10 billion token subset of the FineWeb-Edu dataset (Penedo et al., 2024), a high-quality, carefully curated collection of educational content from the web. Our baseline for comparison are standard full parameter pre-training (referred to as "Full"), where all parameters of the model are trainable and ReLoRA. Importantly, all methods use the exact same underlying model architecture and total number of parameters; the difference lies solely in the number of trainable parameters during training. All models are based on the Qwen2.5 architecture (Qwen et al., 2025): in addition to the standard 0.5B and 1.5B parameter versions, we developed a custom 100M parameter variant by scaling down the same architecture. Detailed specifications of the model architectures are provided in Table 6. To summarize the parameter reduction achieved through BPT in Table 1. As the model scales, the number of trainable parameters required to match the full parameterized model reduces drastically since the number of trainable parameters for layers like the embedding layer does not have an out sized effect on larger models. We also additionally pre-train OLMo-2-1B (OLMo et al., 2024) on the same dataset. Although both models are transformer-based auto-regressive language models, they differ slightly in their attention implementation (Grouped Query Attention vs Multi-head Attention).

Additionally, Qwen-2.5 ties its input and output embeddings, whereas OLMo-2 has separate input and output embeddings. Further details on hyper-parameters and experimental setup can be found in Appendix.

Table 1: Trainable parameter counts by method and model size

| Model | Full Trainable | BPT Trainable | Reduction% | Reduction% (non-embed) |
|---|---|---|---|---|
| Small | 108M | 89M | 17.8 | 63.5 |
| Medium | 0.5B | 0.2B | 58.2 | 80.4 |
| Large | 1.5B | 0.5B | 65.9 | 77.4 |

For a fair comparison, all experiments for a given model size used the same batch size and were trained for the same total number of steps. The optimizer used was AdamW (Loshchilov & Hutter, 2019), and the learning rate followed a warm-up and cosine decay schedule (WSD).

## 5.1 PRE-TRAINING PERFORMANCE AND PARAMETER EFFICIENCY

Our pre-training experiments demonstrate that BPT successfully matches the performance of full parameter training across different model scales while achieving substantial reductions in trainable parameters. Figure 1 summarizes our experimental results across different model sizes. The x-axis represents the trainable parameters not including embedding parameters. We show for a given model size that BPT reaches similar log-perplexity performance in a given validation set as the full parameterized model with significantly fewer trainable parameters.

Figure 3 compares the eval log perplexity on the validation set for the Qwen-2.5-1.5B and OLMo-2 between the Full, ReLoRA and BPT. We show that even with **66% less trainable parameters**, we achieve the same eval loss. The log-perplexity of validation data shows that the model does not simply overfit the training data and has the similar evaluation loss profile as the baseline. Both BPT and ReLoRA has almost identical number of trainable parameters, however ReLoRA eval loss slowly starts to diverge from that of BPT even though both method use similar high learning rate.

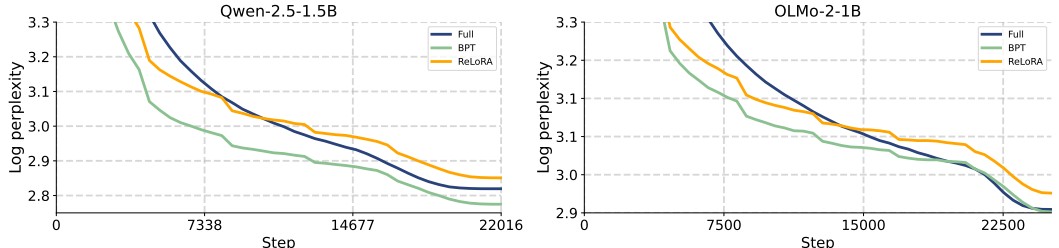

Figure 3: Eval loss over training steps. Left: Qwen-2.5-1.5B. Right: OLMo-2-1B. BPT tracks the Full baseline closely and outperforms ReLoRA over time, despite using substantially fewer trainable parameters

## 5.2 DOWNSTREAM EVALUATION

The training loss and evaluation loss indicate strong performance for our proposed methods. To further evaluate whether the parameter efficiency of BPT impacts the model's ability to generalize to unseen tasks, we perform downstream evaluations on our Qwen-2.5-1.5B and OLMo-2-1B models. For all our evaluations, we do not further finetune any additional dataset and report zero-shot performance. Evaluations were performed using the lighteval framework (Habib et al., 2023), which is a derivative of the lm evaluation harness. We use 17 different evaluations tasks in total covering different sub-topics of language modeling. We group these 17 tasks into 3 groups:

**Natural Language Understanding** This category includes classic NLU benchmarks, which test a model's ability to perform sentence-level and paragraph-level reasoning. Tasks such as MNLI,

QNLI, RTE, and WNLI focus on recognizing textual entailment and semantic similarity, while SST-2 evaluates sentiment classification. These tasks collectively assess a model's syntactic and semantic comprehension capabilities.

Table 2: Performance on Natural Language Understanding: For all metrics higher is better

| Model | Method | SST-2 | RTE | QNLI | WNLI | MNLI | **Avg** |
|---|---|---|---|---|---|---|---|
| | Full | 51.9 | 54.1 | 49.5 | 42.2 | 33.2 | 46.2 |
| Qwen-2.5-1.5B | ReLoRA | 50.9 | 55.6 | 49.4 | 50.7 | 34.0 | 48.1 |
| | BPT | 49.9 | 55.6 | 49.8 | 46.5 | 34.9 | 47.3 |
| | Full | 50.9 | 55.2 | 49.5 | 43.7 | 35.0 | 46.9 |
| OLMo-2-1B | ReLoRA | 52.3 | 53.4 | 49.5 | 36.6 | 34.8 | 45.3 |
| | BPT | 58.1 | 50.9 | 49.6 | 46.5 | 34.6 | 47.9 |

**Commonsense Reasoning** Commonsense reasoning tasks evaluate a model's grasp of implicit, everyday knowledge and its ability to perform inference beyond surface-level text. This includes HellaSwag (Zellers et al., 2019) and WinoGrande (Sakaguchi et al., 2019), which require plausibility and pronoun disambiguation skills, as well as PIQA(Bisk et al., 2019), WSC, BoolQ(Clark et al., 2019), and WiC (Pilehvar & Camacho-Collados, 2019), which challenge the model's understanding of physical reasoning, coreference, binary question answering, and word sense disambiguation, respectively. Table 3 shows the results for this category.

Table 3: Performance on Commonsense Reasoning Tasks: For all metrics higher is better

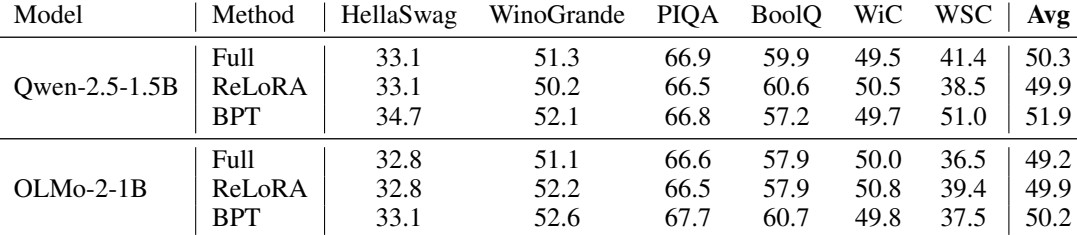

| Model | Method | HellaSwag | WinoGrande | PIQA | BoolQ | WiC | WSC | **Avg** |
|---|---|---|---|---|---|---|---|---|
| | Full | 33.1 | 51.3 | 66.9 | 59.9 | 49.5 | 41.4 | 50.3 |
| Qwen-2.5-1.5B | ReLoRA | 33.1 | 50.2 | 66.5 | 60.6 | 50.5 | 38.5 | 49.9 |
| | BPT | 34.7 | 52.1 | 66.8 | 57.2 | 49.7 | 51.0 | 51.9 |
| | Full | 32.8 | 51.1 | 66.6 | 57.9 | 50.0 | 36.5 | 49.2 |
| OLMo-2-1B | ReLoRA | 32.8 | 52.2 | 66.5 | 57.9 | 50.8 | 39.4 | 49.9 |
| | BPT | 33.1 | 52.6 | 67.7 | 60.7 | 49.8 | 37.5 | 50.2 |

**Reading Comprehension / QA** assesses the model's ability to answer questions based on short passages or structured knowledge, measuring the model's ability to extract, synthesize, and reason over textual information. ARC easy (ARC-E), ARC challenge (ARC-C) (Clark et al., 2018), OpenBookQA (OBQA) (Mihaylov et al., 2018), and SciQ (Johannes Welbl, 2017) test scientific and factual reasoning, while MultiRC (Khashabi et al., 2018) requires identifying multiple correct answers based on context, further testing multi-sentence comprehension. MMLU (Hendrycks et al., 2021) evaluates broad subject-area knowledge and reasoning across disciplines such as mathematics, law, and medicine.

Table 4: Performance on QA and Reading Comprehension: For all metrics higher is better

| Model | Method | SciQ | OBQA | ARC-E | ARC-C | MultiRC | **Avg** |
|---|---|---|---|---|---|---|---|
| | Full | 81.1 | 23.6 | 59.2 | 23.0 | 48.9 | 47.2 |
| Qwen-2.5-1.5B | ReLoRA | 82.6 | 23.2 | 58.8 | 24.5 | 42.8 | 46.4 |
| | BPT | 81.2 | 23.0 | 61.9 | 27.3 | 48.6 | 48.4 |
| | Full | 82.8 | 23.2 | 58.9 | 23.5 | 43.2 | 46.3 |
| OLMo-2-1B | ReLoRA | 83.7 | 23.4 | 58.8 | 25.4 | 45.7 | 47.4 |
| | BPT | 82.2 | 23.6 | 59.9 | 25.7 | 44.3 | 47.1 |

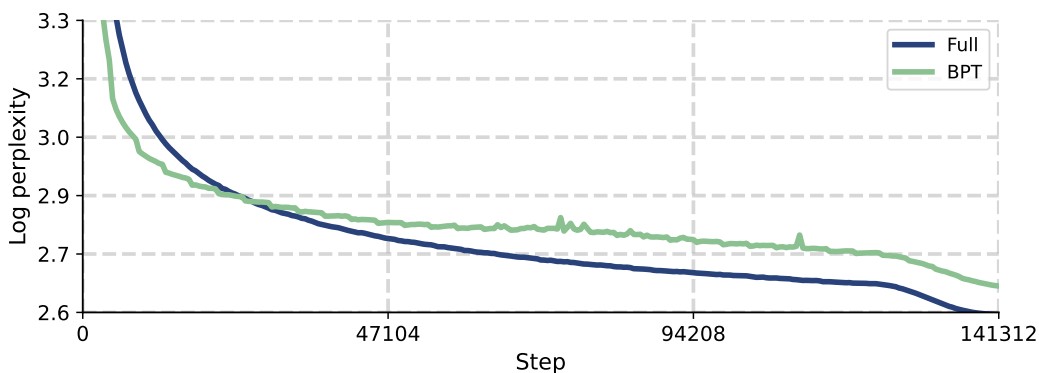

Figure 4: Qwen-2.5-1.5B: Validation Loss vs. Training Steps (Full vs. BPT) for 60B tokens

## 5.3 LONGER TRAINING

To further assess the effectiveness of BPT, we pre-train Qwen-2.5-1.5B model with 60 billion tokens of fineweb-edu. With the same setup as our main experiments. The final eval difference between BPT and Full is only 0.05 log perplexity with 66% fewer trainable parameters.

## 5.4 BASELINE COMPARISON OF PEFT METHODS

To evaluate the performance of PEFT methods such as LoRA and HyperAdapt, we pre-train the Qwen-2.5-0.5B model in the same setup as our main experiments with 10 billion tokens and report their eval loss along with the trainable parameters.

The result is summarized in Table 5. HyperAdapt, which only uses diagonal matrices to update W, performs poorly since the base weight matrix W does not contain any relevant subspace that can be adjusted, since W at initialization is just a random matrix. LoRA without periodic rank accumulation performs better than HyperAdapt, and the difference between LoRA's eval loss and ReLoRA's eval loss is only 0.01. BPT and Full both have eval that are similar compared to other methods.

## 6 CONCLUSION

In this paper, we introduced Bi-Phase Training (BPT), a parameter-efficient pre-training method that couples constrained high-rank transformations through diagonal matrices and explores low-dimensional subspaces via low-rank matrices. The resulting update preserves high-rank expressivity while drastically reducing the number of trainable parameters. We prove an upper bound of $2R + r$ on the update rank of BPT and describe a compounding mechanism in which periodic merges of UV into W continuously expand the accessible subspace, while diagonals adaptively rescale accumulated directions. Empirically, across models from 100M to 1.5B parameters, BPT tracks the fully parameterized baseline during pre-training and, at 1.5B scale, matches validation loss with 66% fewer trainable parameters. Zero-shot results on 17 diverse downstream tasks show that these savings do not come at the cost of generalization. BPT offers a practical path to scaling foundation-model pre-training while being efficient.

Table 5: Comparison of PEFT Methods: Trainable Parameter and Eval Loss

| Method | Trainable Param | Eval Loss |
|---|---|---|
| Full | 494M | 3.02 |
| HyperAdapt | 137M | 5.49 |
| LoRA | 206M | 3.05 |
| ReLoRA | 206M | 3.04 |
| BPT | 206M | 3.01 |

**Limitations** In this work, we only investigated the application of our method in language modeling. We leave extending Bi-Phase Training (BPT) to other domains—such as computer vision, diffusion models, and broader deep learning applications for future work.

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

## A  MODEL ARCHITECTURE

Table 6: Model Architecture Detail and Total Training Tokens

| Models | Layers | Heads (Q / KV) | Hidden Size | Intermediate Size | Training Tokens |
|--------|--------|----------------|-------------|-------------------|-----------------|
| 100M | 8 | 8/ 2 | 512 | 2048 | 10B |
| 0.5B | 24 | 14 / 2 | 896 | 4864 | 10B |
| 1.5B | 36 | 16 / 2 | 2048 | 11008 | 10B |

## B  HYPERPARAMETERS AND EXPERIMENTAL SETUP

For BPT and other baselines, we applied this method exclusively to the linear layers within the transformer blocks. Parameters such as layer normalization, embedding layers, and (lm_head) were kept fully trainable. For the Qwen2.5 architecture, attention matrices, such as the query projection layer `q_head`, have dimensions 518 by 128. Applying BPT to such layers would increase the number of trainable parameters if the low-rank matrices' rank is set to 128, undermining the parameter efficiency goal for these specific matrices. Hence, we leave them unchanged. To ensure that our method is robust across model architectures.

We did not conduct extensive hyperparameter search for learning rate. Following the trend of PEFT papers, where learning rate is higher compared to full fine-tuning. We simply used 10x of the base learning rate which was 2e-4 for all our base models. For ReLoRA with Qwen 1.5B model and LoRA with Qwen 0.5B model, we used a learning rate of 1e-3 after multiple crashes with learning rate 2e-3. For all our experiments, we used Warmup-Stable-Decay (WSD) learning rate scheduler. Additionally, all the parameters in the model were trained using bf16 including the low-rank matrices with the exception of the diagonal matrices which were kept in fp32 for stability.

Table 7: The hyperparameters used for pre-training Qwen-2.5 Models.

| Method | Models | 100M | 0.5B | 1.5B |
|---|---|---|---|---|
| | Optimizer | | AdamW | |
| | Warmup Steps | | 2000 | |
| | Decay Steps | | 10000,8000,8000 | |
| | Max Grad Norm | | 1.0 | |
| | Max Seq. Len | | 512 | |
| | Batch Size | | 320,512,1024 | |
| | LR Schedule | | WSD | |
| | Tokens | | 10B | |
| Full | Learning Rate | | 2e-4 | |
| BPT | Learning Rate | | 2e-3 | |
| | Rank | | 128,128,256 | |
| ReLoRA | Learning Rate | | 2e-3,1e-3 | |
| | Rank | | ,128,256 | |
| LoRA | Learning Rate | | 1e-3 | |
| | Rank | | 128 | |
| HyperAdapt | Learning Rate | | 2e-3 | |

Table 8: The hyperparameters used for pre-training OLMo-2-1B.

| Method | Models | 1B |
|---|---|---|
| | Optimizer | AdamW |
| | Warmup Steps | 2000 |
| | Decay Steps | 8000 |
| | Max Grad Norm | 1.0 |
| | Max Seq. Len | 512 |
| | Batch Size | 960 |
| | LR Schedule | WSD |
| | Tokens | 10B |
| Full | Learning Rate | 2e-4 |
| BPT | Learning Rate | 2e-3 |
| | Rank | 256 |
| ReLoRA | Learning Rate | 2e-3 |
| | Rank | 256 |