# OpenReview forum: "Bi-Phase Training: Learning Efficiently in High Dimensions"
_ICLR.cc/2026/Conference — ICLR 2026 Conference Withdrawn Submission_

### Official Review · Reviewer_jAWY · 2025-10-28

**Soundness:** 3
**Presentation:** 3
**Contribution:** 2
**Rating:** 4
**Confidence:** 3

**Summary:**

This paper propose BPT (Figure 2), a parameter-efficient pretraining method by introducing trainable low-rank factors and diagonal factors. Specifically, similar to ReLoRA, BPT leverages trainable low-rank factors, which are refreshed and accumlated to the full size weight every N steps. The core innovation of BPT is that it improves the expressiveness of the trainable low-rank factors by wrapping the weight update within two additional trainable diagonal matrices.

**Strengths:**

- The authors improves the expressiveness of ReLoRA with moderate extra trainable parameters.
- The authors provide an upper bound of the rank of the update.
- The authors evaluate the performance of BPT on natural language understanding, commonsense reasoning, and reading comprehension tasks.

**Weaknesses:**

**Concern 1. The performance gain over ReLoRA is inconsistent.**

As shown in Table 2, Table 3, and Table 4. The performance gain of BPT compared to ReLoRA is inconsistent and marginal. From my perspective, BPT is largely built upon ReLoRA with a few extra trainable parameters. I wonder if there are some more significant empirical evidence to support that this small modification is essential enough to overcome some critical issues or bad properties in ReLoRA.


**Concern 2. Empirical evidence for compounding effect in Section 3.1.**

To my understanding, the most essential difference between BPT and ReLoRA is the compounding effect introduced in Section 3.1. I wonder if the authors can provide more empirical evidence to support this claim. Does ReLoRA fail to achieve the 'useful directions discovery' effect and 'optimization direction improvement' in practice? If so, why are the root causes of this issue and how does the introduced trainable diagonal matrices help to address this issue?


**Concern 3. Comparison with other parameter-efficient methods.**

As shown in [1,2], various parameter-efficient methods have been proposed to improve training performance within low-rank structures. Many of them exhibit better pretrained PPL and memory consumption than ReLoRA. I wonder if the authors can provide a comprehensive comparison with these methods in terms of performance and computational efficiency.

[1]. Zhao, J., Zhang, Z., Chen, B., Wang, Z., Anandkumar, A., & Tian, Y. (2024). Galore: Memory-efficient LLM training by gradient low-rank projection. arXiv preprint arXiv:2403.03507.

[2]. Zhu, H., Zhang, Z., Cong, W., Liu, X., Park, S., Chandra, V., Long, B., Pan, D. Z., Wang, Z., & Lee, J. (2024). APOLLO: SGD-like memory, AdamW-level performance. arXiv preprint arXiv:2412.05270.

**Questions:**

NA

---

### Official Review · Reviewer_Xe6g · 2025-10-28

**Soundness:** 3
**Presentation:** 2
**Contribution:** 2
**Rating:** 2
**Confidence:** 5

**Summary:**

This paper introduces Bi-Phase Training (BPT), a parameter-efficient approach to pre-training large-scale foundation models. The method combines high-rank transformations using diagonal matrices with low-rank matrix updates to reduce the number of trainable parameters while preserving the performance of fully parameterized models. The authors empirically demonstrate that BPT achieves comparable performance to fully trainable models across a range of model sizes and tasks, with up to 66% fewer trainable parameters. The paper also provides theoretical insights into the rank dynamics of the proposed method and validates its efficacy on 17 diverse downstream tasks.

**Strengths:**

1. The paper tackles an important and timely problem in the field—reducing the computational cost of training large models—making it potentially impactful if executed well.
2. The authors rigorously evaluate BPT across multiple scales (100M to 1.5B parameters) and demonstrate its comparable performance on 17 downstream tasks, including natural language understanding, commonsense reasoning, and reading comprehension.

**Weaknesses:**

1. The method is largely incremental, building directly on existing approaches like LoRA and ReLoRA. While BPT introduces diagonal matrix scaling, this addition feels more like an extension rather than a fundamentally new idea. The lack of ablation studies to isolate the impact of the diagonal matrices versus low-rank updates further weakens the claim of novelty.
2. The experiments focus exclusively on language modeling tasks, leaving open questions about whether the method generalizes to other domains like computer vision, speech, or generative models. The authors themselves acknowledge this limitation, but it significantly reduces the broader impact of the work.
3. The main experiment only compared the full parameter with ReLORA, without comparisons with methods such as HyperAdapt and LoRA.
4. The paper emphasizes parameter reduction but does not provide sufficient analysis of computational trade-offs. For example, the impact on training time, convergence speed, and memory overhead is not discussed. This omission makes it difficult to assess the practical value of BPT.

**Questions:**

1. Could you provide a more detailed explanation of how BPT fundamentally differs from prior methods such as LoRA, ReLoRA, and HyperAdapt? Specifically, what is the unique advantage of combining diagonal matrices with low-rank updates, and why is this combination necessary?
2. While BPT reduces trainable parameters, how does it impact training time, convergence speed, and memory usage? Does the introduction of diagonal matrices and periodic merging introduce additional computational overhead?

---

### Official Review · Reviewer_u8cQ · 2025-10-30

**Soundness:** 2
**Presentation:** 3
**Contribution:** 2
**Rating:** 4
**Confidence:** 4

**Summary:**

This paper proposes a new parameter-efficient approach for pre-training LLMs, which builds on prior parameter efficient finetuning methods such as LoRA and ReLoRA. They argue that low-rank update alone may be too restrictive for pre-training from scratch. So they update the weight metrix by \Delta W = A (W + UV) B - W where A and B are trainable diagonal matrices and U and V are low-rank matrics. SImilar to ReLoRA, they periodically merge W + UV -> W and reinitalize U and V again to explore new subspaces. They show a theoretical bound for \Delta W that can achieve high rank updates. Their experimental results on pretraining different sizes of LLM from scratch show that their proposed method achieves similiar or even lower perplexity than full parameter finetuning, and it achieve better downstream task performance than finetuning with ReLoRA.

**Strengths:**

1. This paper addresses paramter-efficient pre-training, which is less explored by previous study.
2. They provide theoretical upper bound to show the expressiveness of the proposed approach.
3. Empirical results on pretraining on different scales and tasks is non-trivial.

**Weaknesses:**

1. The update is a composition of known ingredients such as diagonal scaling plus low-rank plus merging. The theory provided is not showing the convergence of optimization or generalization.
2. Parameter efficiency makes sense for finetuning where compute and data are limited, but the motivation is less clear why we need to contraint parameter updates during pretraining, given that the scaling laws tie performance to total compute and parameter count.
3. The paper emphasizes on fewer trainable params used in training but does not quantify FLOPS, wall time and actual memory used at training. The periodic merge overhead should also be analyzed.

**Questions:**

See weaknesses.

---

### Official Review · Reviewer_QwS4 · 2025-10-31

**Soundness:** 1
**Presentation:** 2
**Contribution:** 1
**Rating:** 0
**Confidence:** 5

**Summary:**

This paper introduces Bi-Phase Training (BPT), a parameter-efficient pre-training method that combines high-rank diagonal transformations with low-rank subspace exploration. The goal is to match the performance of fully parameterized models while substantially reducing the number of trainable parameters.
The authors show that BPT achieves comparable pre-training and downstream performance in some settings with up to 66% fewer trainable parameters for a 1.5B model, demonstrating this on Qwen2.5 and OLMo architectures.

**Strengths:**

- they present an update rule that maintains high-rank capacity with few trainable parameters.
- The writing is clear, making the paper easy to follow.
- Preliminary experiments show some early-stage promise, suggesting the idea could be worth exploring further, however more comprehensive evaluation is needed.

**Weaknesses:**

* No compute-efficiency report.
The author shows that the number of trainable parameters are reduced in some limited settings. However, while the number of trainable parameters is lower, actual savings in FLOPs or wall-clock time are not measured. This is unclear if this approach actually leads to performance benefits.

* Evaluation tasks are outdated and non-expressive.
The 17 benchmarks largely reflect older GLUE-style tasks, which are no longer discriminative for large models. Stronger and more diverse benchmarks (reasoning, multilingual, math, code, instruction following) would give a clearer picture of generalization of current models. Some of the suggeted becnhmarks are GSM8K, MMLU, MMLU-Pro, Math-500, HUmaneval, MBPP, arc challenge ,... these benchmarks are currently widely used to compare large scale languuage models performances.

* Models not trained to convergence.
The paper stops at early stages (10–60B tokens), far below full pre-training scale. The main results are for after training 22K steps which is rather too early to derive any conclusions. Conclusions about matching full training are therefore speculative. The “0.05 log-perplexity difference” after partial training is not sufficient evidence.

* Missing ablations.
There is no clear ablation showing the contribution of the diagonal matrices vs. low-rank components.

**Questions:**

- How far were the models from convergence when results were reported?
- Can you show validation curves until training plateaus?
- Does BPT’s advantage hold when trained for longer, or does it converge to a higher final loss?
- Have you considered evaluating BPT on more challenging and expressive benchmarks beyond classical GLUE-style tasks—particularly those testing reasoning, math, and code generation (e.g., MMLU, MMLU-Pro, GSM8K, Math-500, RACE, HumanEval, MBPP)? These are now widely used for assessing LLM generalization and reasoning capabilities.
- What are the real compute and memory savings (e.g., GPU hours, FLOPs)?
- Could the diagonal scaling matrices be replaced by structured alternatives (e.g., block-diagonal, banded) to improve expressiveness further?

---

### Note · Authors · 2025-12-04

I have read and agree with the venue's withdrawal policy on behalf of myself and my co-authors.